# Effect of Non-Genetic Factors on Reproduction of Extensive versus Intensive Florida Dairy Goats

**DOI:** 10.3390/vetsci9050219

**Published:** 2022-04-30

**Authors:** Pablo Rodríguez-Hernández, João Simões, Cristina Arce, Cipriano Díaz-Gaona, María Dolores López-Fariñas, Manuel Sánchez-Rodríguez, Vicente Rodríguez-Estévez

**Affiliations:** 1Department of Animal Production, International Agrifood Campus of Excellence (ceiA3), Campus de Rabanales, University of Cordoba (UCO), 14071 Córdoba, Spain; v22rohep@uco.es (P.R.-H.); cristina.arce@uco.es (C.A.); pa2digac@uco.es (C.D.-G.); pa1sarom@uco.es (M.S.-R.); 2Veterinary and Animal Research Centre (CECAV), Associate Laboratory for Animal and Veterinary Sciences (AL4AnimalS), Department of Veterinary Sciences, School of Agricultural and Veterinary Sciences, University of Trás-os-Montes and Alto Douro (UTAD), 5000-801 Vila Real, Portugal; jsimoes@utad.pt; 3ACRIFLOR (Asociación Nacional de Criadores de Ganado Caprino de Raza Florida—National Association of Florida Goat Breeders), Department of Animal Production, University of Cordoba, Campus de Rabanales, 14071 Córdoba, Spain; mdoloreslopez@acriflor.org

**Keywords:** age at first kidding, prolificacy, kidding interval, reproductive rate, production system, dairy goat

## Abstract

The main objective of this study was to evaluate the effect of the production system and other environmental/phenotype factors on age at first kidding (AFK), kidding interval (KI) and prolificacy of 19,772 Florida goats reared between 2000 and 2019 on 49 dairy farms (38 farms intensively managed and 11 extensively managed with grazing). AFK was lower on intensive (490.2 ± 0.9 days; *n* = 13,345) than on extensive farms (511.7 ± 2.5 days; *n* = 2357; *p* < 0.001), and highest during the spring season (533.9 ± 2.7 days; *n* = 1932; *p* < 0.001) in both production systems. The average KI was 355.7 ± 0.4 days, mainly varying according to dry period, kidding season and lactation number and kidding type (*p* < 0.01). A significant interaction between production system, kidding season and dry period was observed with the highest AFK on intensive farms during spring and summer for goats presenting a dry period of up to six months. The overall prolificacy (1.64 ± 0.01) increased in recent years in both systems, and it was affected by the production system, but with different patterns; so, the highest prolificacy of primiparous and multiparous goats was observed on extensive and intensive farms, respectively. Besides that, the prolificacy and other reproductive parameters, such as AFK, significantly increased in the last decade, which could be related to management improvements. Besides that, the existence of inter-annual variations should be considered to compare data between farms and years, and to establish the farms’ objectives according to their production systems and production goals.

## 1. Introduction

Livestock benchmarking is considered a useful technique that enables, by data analysis, the improvement in the productive and reproductive performance of farm animals [1,2]. Both productive and reproductive performance are influenced by genetic and non-genetic factors, determining the maximum progress that can be achieved [3]. In this regard, the evaluation of the influence of each factor on the reproduction performance is crucial. So, in dairy production, the reproductive efficiency has an important impact on the overall profitability of the farms and is considered one of its greatest determinants [4,5].

In recent years, an increasing productive volume of goat milk was registered in Spain, reaching 520 million liters per year in 2019 [6]. Thirty-nine per cent of the Spanish dairy goat population is in Andalusia, where 42.1% of Spanish goat milk is produced. Despite the importance of goats in the Spanish livestock economy with a milk value of € 358 million in 2020 [6], the number of studies focused on reproductive parameters’ evaluation (age at first kidding, kidding interval and prolificacy) of local breeds to generate reference indicators of their reproductive efficiency is considered scarce. These reproductive factors affect the profitability of production systems as well as the productivity of the reproductive goats [7,8].

The Florida breed is an autochthonous dairy goat breed, which comes from the Guadalquivir Valley (Southern Spain). Its origin is based on autochthonous Pyrenean goat herds crossed with some Nubian goats in 1920–1930 [9,10]. Currently, the breed census exceeds 27,000 animals registered in its genealogical herd book, and it is widespread throughout the southern half of the Iberian Peninsula (Spain and Portugal). Most of this breed population in Spain is concentrated in Andalusia and Extremadura, with scarce presence in the rest of the geography [11]. It has a mean milk yield of 611.4 ± 7.8 kg, with 4.9% and 3.4% of fat and protein, respectively, during a productive cycle of 344 days, divided into 261 days of lactation and 83.2 days of dry period [11,12,13]. The average weight of the mature doe is 60 kg [14]. The reproduction of this breed is characterized by a seasonal polyestrus depending on photoperiod, and a weak anoestrus in spring, which allows for obtaining kiddings throughout the year. Traditionally, farmers work with three calving batches: at the end of summer, at the end of autumn, and between the end of winter and beginning of spring. However, many farmers currently try to program four calving batches per year, looking for milk production deseasonalization but maintaining one calving per goat and year [13], while others try to concentrate the maximum milk production in autumn and winter. Bucks are usually reared separately from goats until mating, when they are put together during approximately one month. Goat males also show a libido and sperm quality decrease during spring, although their mingling with does (“buck effect”) is often used to stimulate cycling and improve reproductive efficiency.

An intensification tendency has been described by many authors for different livestock breeds in the last two decades [15,16,17,18,19]. This evolution was observed in different dairy breeds in Spain [20,21] and, specifically, in the Florida breed [22]. The main objective of the present study was to estimate the effect of non-genetic factors on age at first kidding (AFK), length of kidding interval (KI) and prolificacy, and to establish the main indicators of reproductive efficiency according to the extensive and intensive systems in Florida goat herds in order to evaluate and to improve farm management in both production systems.

## 2. Materials and Methods

### 2.1. Animal and Farm Management

This study was conducted using data of 19,772 Florida goats reared between 2000 and 2019 on 49 dairy farms located in the southwest of Spain. The area has a Mediterranean climate, with an annual average rainfall of 500 mm and an average annual temperature of 18 °C [23].

The animals included in the present study were managed under intensive or stabled systems (16,330 goats; 38 farms), where animals were reared indoors without grazing; and under extensive systems (3445 goats; 11 farms), where goats mainly grazed during spring and summer and were intensively managed (fed with feed and fodder) for the rest of the year. Both systems of production and their main characteristics are described by [24].

Intensive systems are characterized by a limited surface where animals have usually access to outdoor pens and distributed in different groups depending on their phase of lactation and the level of production, but their diet is totally provided indoors. In this kind of system, animals are mainly milked in early morning and early afternoon turns. In contrast, extensive farms are based on a natural land use with limited and old infrastructures, where grazing is an important food resource throughout the year. As goats are only housed at night, the milking process is usually carried out in the early morning so the animals can return to the field. In terms of handling, the animals are not usually distributed in different groups depending on the production or the lactation phase. The supplementation is variable depending on the time of the year and the availability of natural resources. From a reproductive point of view, the efficiency of extensive farms is markedly lower than intensive ones, and artificial insemination is not a common practice.

### 2.2. Definitions of Data and Variables

Data were provided by the Association of Florida Goats Breeders (ACRIFLOR), arising from the routine data collection carried out in farms. The total data correspond to 65,535 records, where kidding followed by lactation was the main event (unit). Data from abortion occurrence were removed. In addition to the production system (intensive and extensive), which was previously defined, the different variables evaluated in the present study were defined as follows:

1. Age at first kidding (AFK) was defined as the difference between the first kidding date and the birth date of the goat. As a continuous dependent variable, AFK was conducted in days (model 1). Three classes were defined when AFK was employed as a categorical independent variable: ≥16, 14–15 and 11–13 months old.

2. Kidding interval (KI) was defined as the number of days between two consecutive kiddings. As a continuous dependent variable, KI was studied in days (model 2). Three classes were defined when AFK was employed as a categorical independent variable: 180–273, 274–365 and ≥366 days.

3. Prolificacy was defined as the number of kids born per kidding (a continuous dependent variable), differentiating between prolificacy of primiparous goats (using data only from the first parity; model 3) and prolificacy of multiparous goats (using data from 2nd, 3rd, 4th, 5th and ≥6th parities; model 4). Prolificacy was denominated “type of kidding” when classified into three groups (simple, double and triple or more) and used as a categorical independent variable in model 2.

4. Year and season when goats were born were other variables. Birth years were classified into two periods: 2000–2010 and 2011–2017. Season periods were defined as follows: spring (April, May and June), summer (July, August and September), autumn (October, November and December) and winter (January, February and March). This differentiation aimed to underline the influence of the specialization and modernization in Florida rearing and changes in herd management occurring over the years. Year and season were used as categorical independent variables.

5. Year and season of kidding were also included. According to the previous classification, where the threshold was 2010, the data corresponding to the years of kidding were classified into two periods: 2003–2010 and 2011–2019. Year and season of kidding were used as nominal independent variables.

6. Lactation number was classified as 2, 3, 4, 5 and ≥6 as a categorical independent variable. The first lactation was not used because the lactation number was only employed for KI and P of multiparous goat models, which were built with data from ≥2nd parity.

7. Dry period of the previous lactation was defined as the number of days between the end of a lactation and the following kidding. It is classified, as a categorical independent variable, into four groups: ≤61, 62–122, 123–183 and >183 days (model 2).

### 2.3. Statistical Analysis

Generalized linear models (GLM) were made for each reproductive parameter evaluation.

The model 1 for the analysis of AFK records was:
Y_ijkl_ = µ + S_i_ + G_j_ + B_k_ + (S × G)_ij_ + (S × B)_ik_ + (G × B)_jk_ + (S × G × B)_ijk_ + e_ijkl_
where Y_ijkl_ = age at first kidding (days);
µ = population mean;
S_i_ = fixed effect of the production system (two levels; _i_ = intensive, extensive);
G_j_ = fixed effect of the birth year (two levels; _j_ = 2000–2010, 2011–2017);
B_k_ = fixed effect of the birth season (four levels; _k_ = Spring, Summer, Autumn, Winter);
(S × G)_ij_, (S × B)_ik_ and (G × B)_jk_ = two-way interaction effects between S, G and B factors;
(S × G × B)_ijk_ = S × G × B is three-way interaction effect;
_eijkl_ = experimental error normally and independently distributed with zero mean and variance σ^2^.

The model 2 for the analysis of KI for multiparous goat records was:Y_ijklmny_ = µ + S_i_ + R_j_ + P_k_ + O_l_ + G_m_ + B_n_ + (S × R)_ij_ + … + (G × B)_mn_ + (S × R × P)_ijk_ + … + (O × G × B)_lmn_ + e_ijklmny_
where Y_ijklmny_ = kidding interval (days);
µ = population mean;
S_i_ = fixed effect of the production system (two levels; _i_ = intensive, extensive);
R_j_ = fixed effect of the kidding year (two levels; _j_ = 2003–2010, 2011–2019);
P_k_ = fixed effect of the kidding season (four levels; _k_ = Spring, Summer, Autumn, Winter);
O_l_ = fixed effect of the dry period of the previous lactation (four levels; _l_ = ≤61, 62–122, 123–183, >183 days);
G_m_ = fixed effect of the lactation number (five levels; _m_ = 2, 3, 4, 5, ≥6th lactation);
B_n_ = fixed effect of the kidding type (three levels; _y_ = simple, double, triple or more);
(S × R)_ij_ + … + (G × B)_mn_ = two-way interactions for S … B factors;
(S × R × P)_ijk_ + … + (O × G × B)_lmn_ = three-way interactions for S … B factors;
e_ijklmny_ = experimental error normally and independently distributed with zero mean and variance σ^2^.

The model 3 for the analysis of prolificacy on primiparous goat records was:Y_ijklm_ = µ + S_i_ + G_j_ + F_k_ + B_l_ + (S × G)_ij_ + (S × F)_ij_ + (S × B)_il_ + (G × F)_jk_ + (G × F)_jl_ + (F × B)_kl_ + (S × G × F)_ijk_ + (S × G × B)_ijl_ + (G × F × B)_jkl_ + e_ijklm_
where Y_ijklm_ = prolificacy;
µ = population mean;
S_i_ = fixed effect of the production system (two levels; _i_ = intensive, extensive);
G_j_ = fixed effect of kidding year (two levels; _j_ = 2003–2010, 2011–2019);
F_k_ = fixed effect of kidding season (four levels; _k_ = Spring, Summer, Autumn, Winter);
B_j_ = fixed effect of AFK (three levels; _l_ = 11–13,14–15, ≥16 months);
(S × G)_ij_, (S × F)_ij_, (S × B)_il_, (G × F)_jk_, (G × F)_jl_ and (F × B)_kl_ = two-way interactions for A, G, F and B factors;
(S × G × F)_ijk_, (S × G × B)_ijl_ and (G × F × B)_jkl_ = three-way interactions for A, G, F and B factors;
e_ijklm_ = experimental error normally and independently distributed with zero mean and variance σ^2^.

The model 4 for the analysis of prolificacy on multiparous goat records was:Y_ijklmn_ = µ + S_i_ + R_j_ + P_k_ + G_l_ + F_m_ + (S × R)_ij_ + … + (G × F)_lm_ + (S × R × P)_ijk_ + … + (P × G × F)_klm_ + e_ijklmn_
where Y_ijklmn_ = prolificacy;
µ = population mean;
S_i_ = fixed effect of the production system (two levels; _i_ = intensive, extensive);
R_j_ = fixed effect of the kidding year (two levels; _j_ = 2003–2010, 2011–2019);
P_k_ = fixed effect of the kidding season (four levels; _k_ = Spring, Summer, Autumn, Winter);
G_l_ = fixed effect of the lactation number (five levels; _l_ = 2, 3,4, 5, ≥6th lactation);
F_m_ = fixed effect of the KI (three levels; *_n_* = ≤ 150, 151–305, ≥306 days);
(S × R)_ij_ + … + (G × F)_lm_ = two-way interactions for S … F factors;
(S × R × P)_ijk_ + … + (P × G × F)_klm_ = three-way interactions for S … F factors;
e_ijklmn_ = experimental error normally and independently distributed with zero mean and variance σ^2^.

The JMP14 program from SAS was used for the evaluations, and the Tukey test was used to evaluate differences between pairs. Differences between groups and their interactions were considered significant at the 0.05 level. The results are shown as mean ± standard error (±SE).

## 3. Results

The *p* values of each model and all the respective effects (from factors and factor interactions) are reported in the Appendix A. Only the most significant or important effects for the practice management interaction are addressed here.

### 3.1. Age at First Kidding

The average AFK of Florida goats was 493.4 ± 0.8 days (*n* = 15,702; 95% confidence interval, 95% CI: 491.8–495.1 days).

Overall, AFK was significantly affected by the three factors studied: it was lower in intensive (490.2 ± 0.9 days; *n* = 13,345) than in extensive systems (511.7 ± 2.5 days; *n* = 2357; *p* < 0.001). An important reduction in AFK from the 2000–2010 (509.2 ± 1.2 days; *n* = 7454) to 2011–2017 period (479.2 ± 1.1 days; *n* = 8248; *p* < 0.001) was observed. AFK was significantly higher in spring (533.9 ± 2.7 days; *n* = 1932; *p* < 0.001) than in summer (490.4 ± 1.6 days; *n* = 3112), autumn (483.5 ± 1.3 days; *n* = 5508) and winter (490.8 ± 1.7 days; *n* = 55,150). In addition, the general pattern of differences persisted between both production systems according to two- and three-way factor interactions (Table 1).

### 3.2. Kidding Interval

The average KI of Florida goats was 355.7 ± 0.4 days (*n* = 43,546; 95% CI: 354.9–356.5 days). KI was significantly influenced by dry period, kidding season, lactation number and kidding type (*p* < 0.01; Figure 1), but not by production system (*p* = 0.53) or kidding year period (*p* = 0.23). Nevertheless, a significant three-way interaction between production system, kidding season and dry period was observed (*p* < 0.001; Table 2); as a consequence, during spring and summer, a higher KI was observed on intensive farms than on extensive farms, but only for shorter dry periods (≤61 and 62–122 days).

Moreover, the lactation number significantly (*p* < 0.001) interacted with kidding season, dry period and both variables, simultaneously. Overall, the highest KI was observed in goats after the second lactation in all kidding seasons, except during spring (lactation number × kidding season: *p* < 0.001); in this last case, goats at second lactation had a higher KI (389.8 ± 1.7 days; *n* = 2412; *p* < 0.001) than goats at third (379.0 ± 2.2 days; *n* = 1983), fourth (376.1 ± 2.7 days; *n* = 1206), fifth (369.5 ± 3.4 days; *n* = 764) or ≥sixth lactation (363.4 ± 4.1 days; *n* = 701).

### 3.3. Prolificacy in Primiparous Goats

The mean prolificacy of Florida goats was 1.64 kids (95% IC: 1.64–165; *n* = 104,078), with an average prolificacy of 1.39 ± 0.01 kids (*n* = 15,134; 95% CI: 1.38–1.40) in primiparous goats.

Overall, primiparous goat prolificacy was affected by all the factors studied (*p* < 0.01) except kidding season (*p* = 0.12). Nevertheless, the interactions between seasons and all the three remaining factors were significant (Table 3). Their prolificacy increased from 2003–2010 (1.28 ± 0.01; *n* = 5459) to 2011–2019 (1.45 ± 0.01; *n* = 9675; *p* < 0.001). Considering combined year periods, prolificacy was lower in extensive (1.28 ± 0.01; *n* = 2260) than in intensive systems (1.41 ± 0.01; *n* = 12,874; *p* < 0.001).

Prolificacy of primiparous goats increased as AFK increased (*p* = 0.001): prolificacy was 1.28 ± 0.01 (*n* = 32,456), 1.40 ± 0.01 (*n* = 5947) and 1.44 ± 0.01 (*n* = 5931) for 11–13, 14–15 and ≥16 months, respectively. The interaction of AFK with kidding season (*p* < 0.001) played an important role in prolificacy variation; no differences of prolificacy between seasons were observed for the 11–13 and ≥ 16 months AFK group. In contrast, in the 14–15 months AFK group, prolificacy was higher in autumn (1.47 ± 0.01; *n* = 1490) than in winter (1.38 ± 0.01; *n* = 2698; *p* < 0.05) or spring (1.35 ± 0.01; *n* = 1562; *p* < 0.001).

### 3.4. Prolificacy in Multiparous Goats

The average prolificacy in multiparous Florida goats was 1.75 ± 0.01 kids (n = 40,062; 95% CI: 1.74–1.75). Lactation number and KI strongly affected prolificacy in multiparous goats (*p* < 0.001; Figure 2), followed by the production system (*p* < 0.01). Overall, prolificacy obtained for extensive systems was significantly higher than in intensive ones: 1.78 ± 0.01 (*n* = 6822) and 1.74 ± 0.01(*n* = 33,240; *p* < 0.001), respectively. In addition, a significant prolificacy increase period was detected throughout the study: from 1.73 ± 0.01 (*n* = 9910) to 1.75 ± 0.01 (*n* = 30,152; *p* < 0.001) for the 2003–2010 and 2011–2019 periods, respectively.

Nevertheless, a significant production system × kidding year interaction was observed (*p* < 0.001). As a consequence, the increase in prolificacy between the 2003–2010 and 2011–2019 periods was mainly observed on extensive farms (1.59 ± 0.02 and 1.82 ± 0.01, respectively; *p* < 0.001) than on intensive farms (1.73 ± 0.01 and 1.74 ± 0.01, respectively; *p* = 0.40) (Table 4). No significant interactions were observed between production system and kidding interval (*p* = 0.49) or between production system, kidding interval and kidding year (*p* = 0.71).

## 4. Discussion

### 4.1. Age at First Kidding

The average AFK of Florida goats (493.4 ± 0.8 days; about 16.5 months) was similar to some other goat breeds (Table 5) but can vary for several reasons, as the present study shows. Overall, Florida goats reared under intensive systems present a shorter AFK (21.5 days) than goats reared on extensive farms. This finding reflects the different management strategies between intensive and extensive systems and its influence, especially in feed management, as [24] describes. In this regard, AFK was found to be largely dependent on an adequate nutritional regime [25]. Furthermore, [26] reported that the nutritional requirement and rearing environment can considerably affect the reproductive performance. Under intensive production systems, deficiencies in goat rearing are usually overcome; such an environmental factor influence and the planification to obtain the first kidding at 1 year of age are well documented [27].

Moreover, AFK differences between breeds may be related to low weight and body condition of the growing goat, as an inadequate management during the breeding and rearing stages of the animals prolongs the puberty beginning [39]. It could also be due to a mishandling of young goats’ mating, which are usually mated with all the females of the herd and at the same time [14], mainly where artificial insemination is not used. Some authors point out that young goats are mated when they reach 60–70% of mature goats’ live weight [12,40]. The consideration of factors that may be affecting this indicator of reproductive efficiency of goat herds is important, because, if young goats reach sexual maturity as soon as possible, these will have their first birth earlier. Thereby, they increase the possibility of having the largest number of lactations through their productive life and, consequently, lower amortization costs per liter of produced milk [41]. In the current study, a significant decrease of 30 days on AFK between the 2000–2010 and 2010–2017 periods was observed, which is probably related to the modernization and improvement in rearing strategies occurring over these years. Although there are numerous factors that can partly explain yearly variations in goat performance traits, the lower AFK found in the second period could be partially related to farmer decisions depending on feed and milk market circumstances during those years. The feed price increase during 2007 to 2008 [42], and its slow recovery over the following years, led many farmers to look for different handling and feeding strategies in order to reduce costs somehow [24], which is probably related to the lower AFK reported in the 2011–2017 period. Moreover, changes in the goats’ management, on farms where goats are distributed in reproductive groups and receive feed supplementation, could explain this pattern.

The birth season was one of the most significant factors in this study affecting AFK. The general pattern is based on a higher AFK in spring, about more than one month, compared with the rest of the seasons in both production systems and year periods. Thus, some goat breeds have a seasonal variation in reproductive activity throughout the year, exhibiting sexual activity during the autumn and winter, when light to dark ratios are decreasing (shorter days) [43,44,45]. This fact has been described in the Florida breed, where a seasonal anoestrus period affects its reproduction, especially during spring [14]. The higher AFK found in spring is probably related to the fact that goats born in this season reach the reproductive age and weight (between 10 and 12 months) during the months of the year when a decrease in reproductive activity (March-April) has been described, as mentioned above. So, on some farms, where reproductive techniques are not used (e.g., melatonin implants, buck effect, artificial insemination), the goats must go through a waiting period until the adequate time to start cycling, which coincides with the next period when days become shorter again in the northern hemisphere. However, Spanish goat breeds are less seasonal than central European breeds due to their lower latitude [46], which could explain the differences between winter and autumn AFK, expected to be similar because of their shorter days; besides, autumn births correspond to mating under the buck effect during the first half of the year, looking for milk production during autumn and winter [24]. The lowest values were observed in goats born in autumn, because these are mated during the decreasing light period, when they are 9 to 10 months of age, whereas those born between winter and spring must wait longer to enter the reproductive process [12,14].

Nevertheless, in this study, significant AFK differences were observed according to interactions between the factors studied. Considering the two different periods studied, there is a general pattern of a lower AFK in intensive systems than in extensive ones, therefore exhibiting significant seasonal differences (see Table 1). The exception observed during the winter season (+17.4 days) in the period 2011–2017 could be related to the small number of observations from extensive systems. Besides that, during the 2000–2010 period, the differences between both production systems were more pronounced in summer and autumn probably due to the great differences existing between intensive and extensive systems in terms of feeding, equipment and innovation. During the 2011–2017 period, a more specialized rearing and a technification tendency can justify the smaller differences reported between production systems. In this regard, the reproductive management has progressed in removing seasonality and increasing lactation length [24] to achieve continuous production as reported by [14].

According to these results, and for goats’ replacement purposes, while Florida intensive farms should program births to occur during summer and autumn, winter births seem to be the best management option in extensive herds, followed by summer or autumn births. It would be recommendable that farmers improve their breeding and rearing systems to achieve first birth at lower ages, improving growth rates in these phases to mate kids around 10 months, with enough development to guarantee a good first lactation. Reproduction out of season is also recommended to decrease AFK.

### 4.2. Kidding Interval

The average KI of Florida goats (355.7 ± 0.4 days) is higher than the reported KI in indigenous goat breeds mainly at lower latitudes (Table 6). However, this average KI is consistent with [47], who indicate that the target for dairy goat breeds is 365 days: 10 months of lactation and 2 months of dry period.

The KI of Florida goats was mainly affected by dry period, but also kidding season and lactation number played a significant role. Thus, the observed KI increases as the dry period length increases, which constitutes an expected and evident finding considering that KI is divided into lactation and dry periods (see Figure 1). Nevertheless, the prolonging of this dry period is usually due to reproductive failures with longer birth to conception intervals.

The KI pattern regarding the effect of lactation number is consistent with the findings observed by other authors, who reported that KI is affected by several environmental factors, including parity and its subsequent lactation [40,50,51,52]. The lowest KI values (about 340 days) were observed in goats kidding during the summer and autumn seasons, 30–40 days less than goats’ kidding in winter or spring. These differences are probably related to the fact that goats that kid in summer are mated in winter, coinciding with the favorable reproductive period, whereas goats that kid in winter are mated at the beginning of spring, generally using reproductive techniques, thus increasing the KI [14]. In addition, goats kidding in summer often undergo a decrease in milk production in winter being mated in advance and consequently reducing KI. Similar findings were presented in two other Spanish goat breeds: the Payoya goat [53] and Malagueña goat [54]. Besides, the current results exactly match those obtained by [55] in northwestern Croatia, with a mean KI significantly longer in winter and spring than in summer and autumn for Boer goats.

The higher KI observed in the present study after triple kiddings, or kiddings with more kids, than in simple and double births is in agreement with the results obtained by other authors in different goat breeds when comparing single and multiple kids born [56,57], highlighting a lengthening in the KI as the number of kids increases.

According to the production system × kidding season × dry period interaction, goats with short dry periods (≤61 and 62–122 days) had a significantly higher KI on intensive farms than on extensive farms during the spring and summer seasons. In this regard, while high KI has been associated with controlled mating and confinement [58], low KI is generally associated with year-round free mating systems [26]. These differences for short dry periods are also significant but inverse and less clear in autumn, and not significant in winter. As was reported previously, this fact is probably due to the principal aim of producing milk on intensive farms; besides that, on these farms, the food availability and composition can be more stable throughout the year.

Goats kidding in the winter season, which also presented a high KI, were mainly related to unproductive extension (>2 months of dry period), which represents a barrier. In these cases, a better management of feeding and handling would be advisable as well as an improvement in facilities. Nevertheless, it would be necessary to pay special attention to goats kidding in summer and autumn that present shorter kidding intervals.

### 4.3. Prolificacy

The results of prolificacy (mean of 1.64) are similar to other breeds, although there are some differences (Table 7).

Prolificacy significantly increased in the last decade in primiparous and multiparous Florida goats, which is consistent with the improvement in other reproductive parameters over the years, such as AFK. This fact could be related to the differences carried out in goat handling and reproduction, the feeding strategies and the general modernization of dairy goat production, as well as genetic improvement [64].

Prolificacy was also affected by the production system, but with different patterns: for primiparous goats, it was higher on intensive farms during autumn and winter, while for multiparous goats, prolificacy was higher on intensive farms during the first period of year and became higher on extensive farms later in the year. This fact highlights the importance of controlled conditions, which are achieved mainly in intensive systems, for primiparous goats to have acceptable values of prolificacy. It was also noted that while no differences in multiparous goat prolificacy were observed between both periods for intensive systems, there was an important increase in prolificacy for extensive systems (production system × kidding year interaction; *p* < 0.001). Although the main objective of multiparous goats on intensive farms is milk production, this finding suggests that the intensification allowed for obtaining good prolificacy parameters many years ago, equaling prolificacy values in the two periods studied in intensive systems. On the other hand, the improvement in primiparous goat prolificacy in extensive systems underlines the great efforts made in the last decade to improve the dual-purpose production of meat and milk.

In general, the season did not influence prolificacy of primiparous goats but interacted with the remaining studied factors. During the last years (2011–2019 period), prolificacy was higher in kiddings of intensive farms during autumn and winter, and the highest was obtained in summer for both production systems. The optimal growth development of primiparous goats can be achieved when nutritional management is improved, which is usually easier to reach on intensive farms. Instead, in extensive systems, this higher performance of P significantly depends on the environmental conditions, obtaining better results with goat kidding in spring. Besides that, the highest prolificacy was observed for the kidding of primiparous goats at 14–15 months old during autumn than in winter or spring, suggesting that this time threshold is adequate for first kidding age in this breed.

In multiparous goats of extensive farms, the averaged prolificacy was slightly higher in spring and summer during the last years (2011–2019), whereas in intensive farms, this occurred in winter and spring, probably due to food availability and storage according to both production system. The KI progressively increased the prolificacy of multiparous Florida goats, which probably was due to the effect of an early lactation stress on the reproductive system from milk production, while a higher KI allows for proper goat recovery. This seems be particularly important for goats with a low dry period, up to 4 months.

## 5. Conclusions

Overall, AFK was shortened in the last years as well as in intensive production systems, and higher in the spring season. Mating kids around 10 months is feasible and recommendable. The KI average remained stable between the two studied periods, and no influence of the production system was observed. Nevertheless, the seasonal effect was significant, whereas goats kidding in winter presented higher than the KI interval (>2 months of dry period) than in summer or autumn.

The prolificacy average increased in the last years studied and was higher in intensive than extensive production systems, also influenced by the KI in multiparous goats.

The findings of this work could help farmers to handle the factors affecting reproductive performance of their herds properly, establishing the objectives according to their production systems and production goals.

## Figures and Tables

**Figure 1 vetsci-09-00219-f001:**
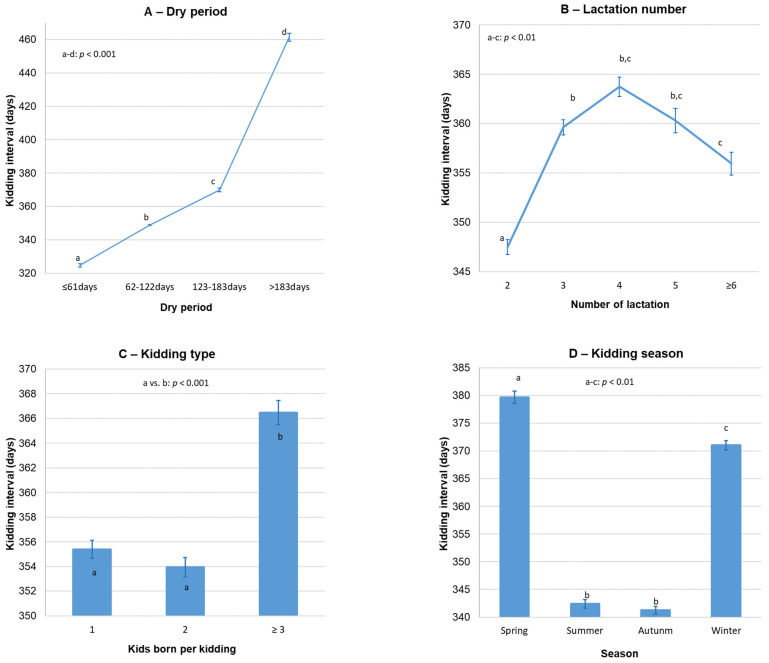
Variation in kidding interval according to dry period (**A**), lactation number (**B**), kidding type (**C**) and kidding season (**D**) of Florida goats. Error bars = ±standard error.

**Figure 2 vetsci-09-00219-f002:**
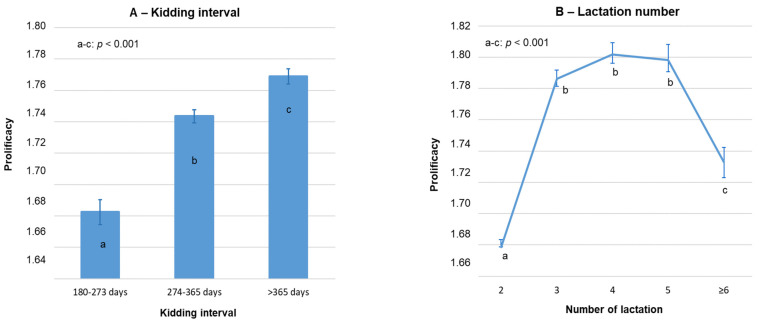
Variation in prolificacy of multiparous Florida goats according to kidding interval (A) and lactation number (**B**). Error bars = ±standard error.

**Table 1 vetsci-09-00219-t001:** Effects of production system, birth season and their interactions on age at first kidding (days ± SE) of Florida goats, and according to each year period.

Year	Birth Season	Production System		*p* Value
		Extensive	Intensive	
**2000–2010**	Spring	562.1 ± 10.0 ^a,b^ (*n* = 122)	550.3 ± 3.9 ^a^ (*n* = 797)	0.27
	Summer	586.2 ± 5.2 ^a^ (*n* = 305)	504.2 ± 3.2 ^b^ (*n* = 802)	<0.001
	Autumn	536.4 ± 4.6 ^b^ (*n* = 338)	486.8 ± 1.7 ^b^ (*n* = 2457)	<0.001
	Winter	493.1 ± 9.2 ^c^ (*n* = 175)	505.2 ± 2.5 ^c^ (*n* = 2458)	0.20
**2011–2017**	Spring	548.8 ± 9.2 ^a^ (*n* = 167)	511.5 ± 4.1 ^a^ (*n* = 846)	<0.001
	Summer	495.5 ± 4.5 ^b^ (*n* = 258)	466.5 ± 2.5 ^b^ (*n* = 1747)	<0.001
	Autumn	485.0 ± 4.1 ^b^ (*n* = 478)	471.0 ± 2.1 ^b^ (*n* = 2135)	0.002
	Winter	461.9 ± 5.7 ^c^ (*n* = 414)	479.3 ± 2.5 ^c^ (*n* = 2103)	0.005

^a,b,c^ superscript letters within columns and for the same year period: *p* < 0.05.

**Table 2 vetsci-09-00219-t002:** Kidding interval (days ± SE) variation according to production system and kidding season for each dry period of Florida goats.

Kidding Season	Dry Period (Days)	Production System		*p* Value
		Extensive	Intensive	
**Spring**	≤61	270.9 ± 7.2 ^a^ (*n* = 169)	337.2 ± 4.6 ^a^ (*n* = 492)	<0.001
	62–122	347.8 ± 6.2 ^b^ (*n* = 328)	376.6 ± 1.2 ^b^ (*n* = 3805)	<0.001
	123–183	353.5 ± 22.7 ^c^ (*n* = 13)	345.9 ± 3.7 ^b^ (*n* = 413)	0.72
	>183	460.5 ± 12.6 ^d^ (*n* = 92)	462.5 ± 5.1 ^c^ (*n* = 652)	0.89
**Summer**	≤61	309.4 ± 3.7 ^a^ (*n* = 482)	332.5 ± 1.7 ^a^ (*n* = 2100)	<0.001
	62–122	333.2 ± 2.0 ^b^ (*n* = 1153)	343.5 ± 0.9 ^b^ (*n* = 5577)	<0.001
	123–183	353.5 ± 22.7 ^a,b^ (*n* = 13)	345.9 ± 3.7 ^b^ (*n* = 413)	0.72
	>183	468.8 ± 20.3 ^c^ (*n* = 57)	443.2 ± 7.4 ^c^ (*n* = 318)	0.19
**Autumn**	≤61	332.5 ± 4.9 ^a^ (*n* = 337)	319.4 ± 1.4 ^a^ (*n* = 2605)	<0.01
	62–122	337.6 ± 1.3 ^a^ (*n* = 2286)	331.4 ± 0.7 ^b^ (*n* = 7172)	<0.001
	123–183	355.3 ± 4.7 ^b^ (*n* = 244)	365.6 ± 2.4 ^c^ (*n* = 1310)	0.09
	>183	445.2 ± 18.3 ^c^ (*n* = 57)	492.1 ± 6.1 ^d^ (*n* = 647)	<0.05
**Winter**	≤61	325.6 ± 7.6 ^a^ (*n* = 150)	333.6 ± 3.5 ^a^ (*n* = 628)	0.35
	62–122	357.0 ± 2.1 ^b^ (*n* = 1337)	361.4 ± 0.9 ^b^ (*n* = 6498)	0.07
	123–183	377.5 ± 4.8 ^c^ (*n* = 263)	371.7 ± 1.9 ^c^ (*n* = 1470)	0.24
	>183	422.3 ± 6.1 ^d^ (*n* = 246)	457.3 ± 4.3 ^d^ (*n* = 1117)	<0.001

^a,b,c^ superscript letters within columns and for the same year period: *p* < 0.05.

**Table 3 vetsci-09-00219-t003:** Effects of production system and kidding season on prolificacy (mean ± SE) of primiparous Florida goats for two periods of years.

Kidding Year	Kidding Season	Production System		*p* Value
		Extensive	Intensive	
**2003–2010**	Spring	1.1 5± 0.03 ^a,b^ (*n* = 164)	1.24 ± 0.01 ^a^ (*n* = 1276)	<0.01
	Summer	1.21 ± 0.09 ^a,b^ (*n* = 36)	1.26 ± 0.02 ^a^ (*n* = 398)	0.62
	Autumn	1.20 ± 0.03 ^a^ (*n* = 177)	1.35 ± 0.02 ^b^ (*n* = 1033)	<0.001
	Winter	1.08 ± 0.02 ^b^ (*n* = 291)	1.30 ± 0.01 ^b^ (*n* = 2084)	<0.001
**2011–2019**	Spring	1.47 ± 0.03 ^a^ (*n* = 222)	1.45 ± 0.01 ^a^ (*n* = 1814)	0.51
	Summer	1.56 ± 0.08 ^a^ (*n* = 71)	1.56 ± 0.02 ^b^ (*n* = 477)	0.96
	Autumn	1.33 ± 0.03 ^b^ (*n* = 388)	1.46 ± 0.01 ^a^ (*n* = 2227)	<0.001
	Winter	1.30 ± 0.02 ^b^ (*n* = 911)	1.48 ± 0.01 ^a^ (*n* = 3565)	<0.001

^a,b^ different superscript letters within columns for each year period: *p* < 0.05.

**Table 4 vetsci-09-00219-t004:** Effects of production system, kidding season and kidding year on prolificacy (mean ± SE) of multiparous Florida goats.

Kidding Year	Kidding Season	Production System		*p* Value
		Extensive	Intensive	
**2003–2010**	Spring	1.49 ± 0.05 ^a^ (*n* = 128)	1.72 ± 0.01 ^a^ (*n* = 1794)	<0.001
	Summer	1.65 ± 0.03 ^b^ (*n* = 279)	1.81 ± 0.02 ^b^ (*n* = 1601)	<0.001
	Autumn	1.60 ± 0.03 ^a,b^ (*n* = 504)	1.77 ± 0.01 ^b^ (*n* = 3119)	<0.001
	Winter	1.54 ± 0.04 ^a,b^ (*n* = 197)	1.67 ± 0.01 ^c^ (*n* = 2288)	<0.01
**2011–2019**	Spring	1.85 ± 0.02 ^a,b^ (*n* = 572)	1.76 ± 0.01 ^a,b^ (*n* = 3778)	<0.001
	Summer	1.89 ± 0.02 ^a^ (*n* = 1242)	1.71 ± 0.01 ^c^ (*n* = 6197)	<0.001
	Autumn	1.80 ± 0.01 ^b,c^ (*n* = 2200)	1.73 ± 0.01 ^b,c^ (*n* = 7796)	<0.001
	Winter	1.77 ± 0.02 ^c^ (*n* = 1700)	1.76 ± 0.01 ^a^ (*n* = 6667)	0.50

^a,b,c^ different superscript letters within columns and for the same year period: *p* < 0.05.

**Table 5 vetsci-09-00219-t005:** Age at first kidding of different local goat breeds.

Goat Breed	Age at First Kidding	Country	Reference
Saanen	16.6 ± 4.4 months	Mexico	[28]
Alpine, La Mancha, Nubian, Oberhasli, Saanen and Toggenburg	16.6 months (507 days)	United States	[29]
Alpine, La Mancha, Nubian, Saanen and Toggenburg	16.9 months (507.9 days)		[30]
Indigenous goats	16 to 18 months	South Africa	[31]
Serrana	15 months	Portugal	[8]
Saanen	13.4 ± 0.1 months	Brazil	[32]
Small East African type	21.0 months (640 days)	Rwanda	[33]
Arsi-Bale	19.2 ± 0.3 months	Ethiopia	[34]
Native goats	13.0 ± 0.4 to 14.3 ± 0.7 months	Bangladesh	[35]
Arab	13.9 ± 1.7 months	Ethiopia	[36]
Oromo	14.9 ± 2.4 months	Ethiopia	[36]
Malagueña	14 months	Spain	[37]
Murciano-Granadina	11–13 months	Spain	[38]

**Table 6 vetsci-09-00219-t006:** Kidding interval data of several local goat breeds.

Goat Breed	Kidding Interval (Days)	Country	Reference
Native goats	261–297	Bangladesh	[35]
Arab	216 ± 54	Ethiopia	[36]
Oromo	234 ± 30	Ethiopia	[36]
Dwarf goats	203.7 ± 46	Pakistan	[48]
Arsi-Bale	280 ± 13.7	Ethiopia	[34]
Small East African type	343	Rwanda	[33]
Alpine, La Mancha, Nubian, Saanen and Toggenburg	379	United States	[49]
	387.4		[30]
Alpine, La Mancha, Nubian, Oberhasli, Saanen and Toggenburg	382		[29]
German fawn	337	Germany	[47]
Korean native goats	207.8 ± 1.8 to 211.6 ± 2.7	Korea	[26]
Murciano-granadina	327	Spain	[38]
Malagueña	290	Spain	[37]

**Table 7 vetsci-09-00219-t007:** Prolificacy of several local goat breeds.

Breed	Prolificacy	Country	Reference
Majorera	2.0 ± 0.03	Spain	[59]
American Alpine	1.9 ± 0.12	United States	[60]
Dairy Crossbred	1.9 ± 0.08		
French Alpine	1.7 ± 0.07		
Nubian	2.0 ± 0.07		
Pygmy	1.9 ± 0.13		
Saanen	1.7 ± 0.11		
Toggenburg	1.6 ± 0.20		
Arsi-Bale	1.6 ± 0.03	Ethiopia	[34]
Malagueña	1.9	Spain	[61]
Murciano-Granadina			
Korean Native goats	1.7 ± 0.03 (extensive groups) 1.8 ± 0.16 (intensive groups)	Korea	[26]
Small East African type	1.75	Ethiopia	[33]
Dwarf goats	1.8 ± 0.8	Pakistan	[48]
Raeini Cashmere	1.1 ± 0.22	Iran	[62]
Markhoz	1.3		[63]

## Data Availability

Not applicable.

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
