# Peer review of "Effect of Non-Genetic Factors on Reproduction of Extensive versus Intensive Florida Dairy Goats"

_vetsci, 2022, doi:10.3390/vetsci9050219_

Round 1

Reviewer 1 Report

This manuscript presents the results of an interesting study on a rather unique breed and has the potential to provide interesting and useful information to the breeders of these goats and the wider reading audience, however there are several issues that need addressing before it could be suitable for publication. 

The first issue is that the reader is provided no information regarding reproductive strategies followed by the goat farms, i.e. are bucks with the does all year or do some farmers practice seasonal breeding strategies, this is particularly important when looking at AFK or KI.

The second issue is with the definition of variables and data, what's the difference between P and type of kidding, they both appear to report the same thing, i.e. kids born per doe, differentiated by parity. Also, why use ranges of variables, i.e. for AFK, why not simply use the actual month of first kidding, it appears as though you have sufficient data observations. If this si not the case then this needs to be explained, similar comments relate to year of kidding and dry period length. 

It would be good to some some standard error bars in figure 1 so the reader can see why the results are/aren't statistically different. You also have two Figure 1s (one at line 196 and one at line 235. The tables need to be inserted in the text so that they align with the text and also the text in the tables should be left justified, not centered. 

Overall, the writing is good, but there are some minor issues with English that could be corrected with a proofread by a qualified English proofreader.

Author Response

Firstly, we thank you for your suggestions and comments. We fully agree and think that these help us to improve the quality of the manuscript. Every change has been marked with blue font.  

Reviewer 2 Report

General comments

Material; and methods

The authors should clearly separate (Lines 98-123) the dependent variables (AFK, KI, P) from the independent variables (factors).

When defining dependent variables, please first provide the full name of the feature, and finally its abbreviation.

The authors used the abbreviation "P" for the fertility trait. In my opinion, this is the wrong approach. The abbreviation "P" generally stands for probability.

The authors wrote: “Generalised linear models (GLM) OR mixed models were made for each reproductive parameter evaluation.” It is not clear which model was used to analyze the individual features?

The authors wrote that they analyzed the features using a mixed model.In my opinion, the fixed effects model was used.

I propose to rewrite the way of presenting the analysis models.

They suggest to prepare a single legend under all models!

I propose to use the Greek letter mi to denote mean population.

Please correct the subscripts at "y" and "e", eg L129 and L133;

L136 and L137, etc.

Results

Maybe I don't understand something, but I didn't find the supplementary file S1?

Personally, I would rather present in tables the mean values for the main factors than for the interactions, for discussion.

The authors give numerical values that I do not have in the tables, lines: 176-183, 199-205, 210-211, 238-241. Please replace the order in which the ranges for 2000-2010 and then 2011-2017 are given in the tables.

In tables, remove the symbol "p" in front of the given p value.

Discussion

The authors cite as many as 64 literature sources. Is it necessary? Shouldn't the number of the cited works be reduced to those presenting the results for goat breeds of a habit similar to Florida goat, utility type, etc.? The work contains as many as 3 tables (Table 5-7) containing the results of other studies. When reading, you get the impression that the article is a review article.

Conclusions

Definitely more synthetic! By removing elements of the summary of the conducted research.

Detailed comments

L36: “These traits” - what are the features? It can be a bit confusing.

L99: “date of birth” – unclear.

L251: In the Discussion chapter, add the numerical values already presented in the Results chapter.

In Table 5, in the column labeled "Age at first Kidding", enter the age in months, and then possibly in days.

L320: Sanchez-Rodrigguez (2008) - change the citation method.

Author Response

(The authors gave the same response as above.)

Round 2

Reviewer 1 Report

The authors are to be congratulated for taking on -board the comments of the reviewers, as a result the newer version of the manuscript is a lot clearer and presents the material well.